# Gold Nanoparticles as Drug Carriers: The Role of Silica and PEG as Surface Coatings in Optimizing Drug Loading

**DOI:** 10.3390/mi14020451

**Published:** 2023-02-15

**Authors:** José Luis Carreón González, Perla Elvia García Casillas, Christian Chapa González

**Affiliations:** 1Grupo de Nanomedicina, Instituto de Ingenieria y Tecnología, Universidad Autónoma de Ciudad Juárez, Avenida del Charro 450, Ciudad Juárez 32310, Mexico; 2Centro de Investigación en Química Aplicada (CIQA), Blvd. Enrique Reyna Hermosillo 140, Saltillo 25294, Mexico

**Keywords:** nanomedicine, nanoparticle, nanoparticle synthesis, gold nanoparticle, drug delivery, drug carrier, polyethylene glycol, silica, nanocarrier, ibuprofen

## Abstract

The use of gold nanoparticles as drug delivery systems has received increasing attention due to their unique properties, such as their high stability and biocompatibility. However, gold nanoparticles have a high affinity for proteins, which can result in their rapid clearance from the body and limited drug loading capabilities. To address these limitations, we coated the gold nanoparticles with silica and PEG, which are known to improve the stability of nanoparticles. The synthesis of the nanoparticles was carried out using a reduction method. The nanoparticles’ size, morphology, and drug loading capacity were also studied. The SEM images showed a spherical and homogeneous morphology; they also showed that the coatings increased the average size of the nanoparticles. The results of this study provide insight into the potential of gold nanoparticles coated with silica and PEG as drug delivery systems. We used ibuprofen as a model drug and found that the highest drug load occurred in PEG-coated nanoparticles and then in silica-coated nanoparticles, while the uncoated nanoparticles had a lower drug loading capacity. The coatings were found to significantly improve the stability and drug load properties of the nanoparticles, making them promising candidates for further development as targeted and controlled release drug delivery systems.

## 1. Introduction

Drug delivery research is an interdisciplinary field that involves the development of new materials, devices, and strategies for delivering therapeutic agents to specific locations in the body. Drug delivery systems are designed to deliver therapeutic agents in order to maximize their effectiveness and minimize their side effects [1]. There are many different types of drug delivery systems, including micro- [2] and nanoparticles [3], nanoplatforms [4,5,6], devices [7,8,9,10], and theragnostic platforms [11,12,13]. These systems can be used to deliver a wide range of drugs, including chemotherapeutics [14,15,16], gene therapies [17,18,19,20], and other biologics [21,22,23,24,25,26]. Optimizing drug loading in drug delivery systems is a critical aspect of their design and performance. This is because the amount of drug loaded onto the delivery vehicle can significantly impact its effectiveness and safety. However, achieving optimal drug loading is not always straightforward, as it can be influenced by a range of physicochemical factors such as the stability of the drug in the delivery system, the size and surface properties of the delivery vehicle, and the interaction between the drug and the delivery vehicle. In addition, the physiological conditions that the delivery system encounters in the body, such as the pH and temperature, can also affect the drug loading. Therefore, optimizing the drug loading requires a thorough understanding of these variables and the ability to carefully control and manipulate them in order to achieve the desired therapeutic effect.

To overcome these limitations, researchers have developed a variety of strategies, such as surface modification or functionalization of drug delivery systems to improve targeting and the use of nanoparticles or nanoplatforms to enhance transport and distribution. Polyethylene glycol (PEG) is a widely used polymer that has many attractive properties for use as a coating in drug delivery systems [27,28]. PEG has been shown to improve the stability of drugs within the delivery system, reduce toxicity [29,30], and enhance the circulation time of the delivery vehicle in the body [31,32]. Additionally, PEG is biocompatible and nonimmunogenic, making it suitable for use in various medical applications. Furthermore, the ability to tailor the size and architecture of PEG coatings allows for fine-tuning the release properties of the drug delivery system. Overall, the use of PEG coatings in drug delivery systems has the potential to significantly improve the safety and efficacy of drug delivery. In the same sense, silica, or silicon dioxide, is a material that has several attractive properties for use as a coating in drug delivery systems. One of the main benefits of silica is its high stability and biocompatibility [33,34,35,36], which makes it suitable for use in a range of medical applications. Additionally, silica is highly porous and has a large surface area, which can be exploited to incorporate a large amount of drug in the delivery system. This can be particularly useful for delivering hydrophilic drugs, which can be difficult to incorporate into other delivery systems. Silica coatings can also be modified to control the release rate of the drug, making it possible to tailor the therapeutic effect. Overall, the use of silica coatings in drug delivery systems offers the potential for the improved stability, bioavailability, and the controlled release of drugs.

Gold nanoparticles are highly attractive for use as drug carriers due to their unique physical and chemical properties, including the ability to be synthesized in various sizes and shapes and easily functionalized with a variety of molecules, which allows for targeted drug delivery [37,38,39,40,41,42]. These features allow gold nanoparticles to target specific cells and tissues and to release drugs in a controlled manner. One of the main challenges in using gold nanoparticles as drug carriers is the need to optimize drug loading. The amount of drug that can be loaded onto the nanoparticles depends on several factors, including the size and shape of the nanoparticles, the surface functionalization, and the physicochemical properties of the drug. Therefore, it is important to carefully optimize the drug loading and release properties of gold nanoparticles to achieve the desired therapeutic effect. Many reports in the literature have demonstrated the potential of gold nanoparticles as carriers for various drugs [43,44,45,46,47,48]; however, more research is needed to optimize the control of drug loading and to compare extensively used coatings such as polyethylene glycol (PEG) [49,50,51,52,53] and silica [54,55,56,57,58].

In addition to optimizing drug loading, it is also important to consider the physicochemical properties of the gold nanoparticles themselves. The size, shape, and surface chemistry of gold nanoparticles can affect their stability, biodistribution, and toxicity. For example, gold nanoparticles coated with polyethylene glycol (PEG) or silica may be more stable in the body and exhibit lower toxicity compared to uncoated nanoparticles. However, PEG and silica coatings may also affect the drug release and targeting properties of the nanoparticles. Therefore, it is important to carefully the effects of different coatings on drug delivery to optimize their use as drug carriers. Further research is needed to fully understand the factors that influence drug loading and release in order to optimize the use of AuNP as drug delivery vehicles. In this study, we synthesized gold nanoparticles and coated them with silica and polyethylene glycol (PEG) to investigate their potential as drug delivery systems. The uncoated gold nanoparticles and the coated nanoparticles were characterized using various techniques and their drug release properties were compared. Our objective is to contribute to the existing literature on AuNP-based biomedical engineering applications by demonstrating the importance of coatings in drug loading. To our knowledge, this is the first time that two of the coatings that have been presented as suitable for therapeutic applications in nanomedicine systems, PEG and silica, have been assessed to compare their drug loading efficiency. Our findings offer a new perspective on the potential of AuNPs in biomedical engineering and pave the way for further studies on their therapeutic efficacy.

## 2. Materials and Methods

The synthesis of gold nanoparticles (AuNPs) was carried out using a reduction method. Chloric Auric Acid (HAuCl_4_) was used as the precursor salt, and sodium citrate was used as the reducing agent. The synthesis process was performed at a critical temperature of 80 °C using platens with stirring function to mix the solutions. To synthesize the AuNPs, 5 mL of HAuCl_4_ was added to 50 mL of distilled water to create solution 1. Solution 2 was prepared by dissolving 115 mg of sodium citrate in 10 mL of distilled water, and the solutions were mixed for 5 min. The resulting mixture was then placed on a stage at 80 °C and agitated at approximately 300 rpm, while the sodium citrate was added to the HAuCl_4_ solution. The nucleation of the nanoparticles was observed by the change in color from transparent to reddish, indicating the formation of the AuNPs. Maintaining the proper temperature during the synthesis process was critical for the successful nucleation of the nanoparticles. The synthesized nanoparticles were characterized using scanning electron microscopy (SEM), and dynamic light scattering (DLS) to confirm their size and size distribution and Fourier transform infrared spectroscopy (FTIR) to determine the functional groups attributable to the coatings.

The method of producing silica-coated gold nanoparticles in our research was inspired by the procedure previously reported in [59], with some modifications. The process involved mixing an ethanolic solution of 4 mM cetyltrimethylammonium bromide (CTAB) and 1 mM tetraethyl orthosilicate (TEOS) with 3 mL of AuNPs and then exposing the mixture to ultrasonic treatment for 4 h. After that, the product was washed multiple times with deionized water and ethanol, collected through centrifugation, and dried at 70 °C for 8 h. This procedure was adapted to produce silica-coated gold nanoparticles for our study. The process for obtaining PEG-coated gold nanoparticles used in our study involved the preparation of a 30 mg/mL solution of PEG 3350 polymer by dissolving 45 mg of the polymer in 1.5 mL of deionized water. The resulting solution was mixed with 3 mL of AuNPs that had undergone prior centrifugation at 8000 rpm. The pH of the solution was then adjusted to between 9 and 10 using 0.1 M NaOH solution. The resulting mixture was subjected to magnetic stirring for 2 h and was dispersed with an ultrasonic homogenizer every 5 min during the first hour and every 15 min during the second hour. The excess PEG not adhered to the surface of the AuNPs was then removed by centrifugation for 30 min at 8000 rpm. This procedure was used as described in the previous literature [60].

The model drug used in this study was ibuprofen, an anti-inflammatory medication commonly used for pain relief and inflammation reduction. Due to its low water solubility, ibuprofen was dissolved in chloroform for purification. Four 400 mg tablets were crushed, and the resulting white powder was weighed to obtain 2.44 g. This powder was added to 200 mL of chloroform in a flask and stirred magnetically until completely dissolved. The solution was filtered to collect the chloroform with ibuprofen, then it was then heated on a hot plate at 60 °C for about an hour to separate the ibuprofen from the chloroform. The resulting white powder was the purified ibuprofen, which was used to load onto the nanoparticles coated with polyethylene glycol (PEG) and silica. To avoid contamination, all the glassware used in the process was washed with acetone and dried in an oven at 60 °C.

The drug loading and release properties of the nanoparticles were evaluated using a UV–Vis spectrophotometer. The results were compared to those obtained with uncoated gold nanoparticles to assess the effect of the coatings on the drug release. The concentration of ibuprofen in the solution was determined using a NANODROP 2000 Spectrophotometer (ThermoFischer, Wilmington, DE, USA). Ibuprofen was dissolved in a 3:1 ratio of acetone to deionized water to improve its solubility, and ten Eppendorf tubes were prepared with different concentrations of the drug. The tubes were incubated, homogenized, and labeled to create a calibration curve and confirm the presence of different concentrations of ibuprofen.

The drug loading onto gold nanoparticles (AuNPs) was performed using a 3:1 ratio solution of acetone and water to dissolve the drug in an aqueous medium. Ten Eppendorf tubes were prepared with different concentrations of the drug-loaded solution by mixing 0.5 mL of the drug solution with 0.5 mL of AuNPs solution. The tubes were incubated for 30 min in a Thermolyne Roto Mix Type 50800 orbital shaker to allow the drug to attach to the AuNPs. The tubes were then centrifuged for 15 min at 13,000 rpm in an Eppendorf Centrifuge 5415 D to precipitate the AuNPs and recover the liquid solution. The liquid solution was analyzed using UV–Vis spectroscopy to determine the drug concentration. The absorbances were compared to the absorbances obtained in the calibration curve of the drug to calculate the amount of drug present in the solution and the amount of drug anchored to the surface of the AuNPs.

## 3. Results and Discussion

In order to determine the shape and size of the AuNPs, a Scanning Electron Microscopy (SEM) study was conducted (Figure 1). The SEM configuration was set to a secondary electron detector, at the same voltage (15.0 KV) and at different magnifications (130 and 600 K) to visualize the AuNPs. The samples placed in the SEM vacuum chamber exhibited a spherical morphology for all the nanostructures observed during the study. The whole sample was also found to be monodispersed, confirmed by DLS (Figure 2) indicating that the AuNPs were suitable for various alternative therapy applications in biomedical engineering. The size of the AuNPs was determined to be in the range of 5 to 60 nm, with larger sizes observed for the AuNPs coated with PEG or Silica.

It is important to note that the coating has the potential to extend the lifetime of the AuNPs in physiological media. The coating provides a longer lifetime and acts as an anchor for the drug on the nanostructure, serving as a vehicle for drug localization. AuNPs smaller than 10 nm are rapidly excreted and metabolized by the liver due to their size, which allows for unrestricted distribution and mobilization through capillaries in the body. By coating the AuNPs with polymers such as PEG or silica, the dimensions of the AuNPs can be increased to above 10 nm, reaching up to 30 nm, which makes this size range optimal as they are not rapidly eliminated by the liver and kidneys. This can be seen in Figure 3 and Figure 4.

Fourier transform infrared spectroscopy (FTIR) was employed to analyze the AuNPs–PEG sample and confirm the presence of polyethylene glycol (PEG) on the surface of the gold nanoparticles (AuNPs). The FTIR spectrum of the AuNPs–PEG sample showed bands between 3350, 3490, 1630, and 1750 cm^−1^, corresponding to the O-H flexion and stretching bonds, respectively. Additional bands were also observed due to the functionalization of the AuNPs with PEG, including C-H stretching bonds between 1180 and 1210 cm^−1^ and C-H flexion bonds between 950 and 1050 cm^−1^. These bands indicate the successful coating of the AuNPs with PEG. Figure 3 shows the FTIR spectrum of the PEG-coated AuNPs. A zoom was applied to the spectrum to better visualize the bands of the PEG functional groups. FTIR was also used to determine the presence of silica on the surface of the AuNPs. The AuNPs–Silica sample showed bands between 3350 and 3490 and 1630 and 1750 cm^−1^ corresponding to the O-H flexion and stretching vibrations, respectively, as well as bands between 1100, 1300, 800, and 1000 cm^−1^ corresponding to the Si-O-Si vibration and Si-O bond vibration, respectively.

Since the samples were in suspension, we performed a thermogravimetric analysis. The thermogravimetric analysis of a sample of AuNPs was conducted to quantify the mass of the sample. The characterization of the AuNPs is shown in the resulting graph, in which the sample had an initial weight loss of 83.12%. The final weight corresponded to the presence of gold in the sample, while the rest was the medium in which the AuNPs sample was suspended. See Figure 4 for the thermogravimetric analysis of the AuNPs.

The calibration curve and the results of the absorbances obtained after incubation [61] can be seen in Figure 5a,b, respectively. (AuNPs, AuNPs–PEG, and AuNPs–silica). In the following graph, it can be seen how ibuprofen was able to adhere to the AuNPs sample; but from the first concentration, the AuNPs became saturated. It is also worth noting that from the first concentration, the drug continued to adhere to the nanostructures, which may be due to the drug anchoring to several layers on the nanostructures. On the other hand, in the AuNPs–PEG sample, there was an increase in the adhesion of the drug to the nanostructures until it reached saturation at a concentration of around 100 µg/µL. Similarly, it can be seen that the drug was able to adhere to the AuNPs–silica sample, anchoring in a greater amount compared to the AuNPs sample but anchoring in a smaller amount compared to the AuNPs–PEG sample. In this sample, saturation was observed from a concentration of around 100 µg/µL. It was observed that all samples ceased to adsorb the drug when an initial drug concentration of around 100 was used. This may be due to saturation of the drug adsorption sites on the surface of the nanocarriers, leading to an inability to further incorporate the drug. Alternatively, the high drug concentration may result in a decrease in the surface charge density of the nanocarriers, leading to decreased drug–nanocarrier interactions and reduced adsorption. Further investigation is needed to fully understand the mechanisms behind this phenomenon.

The results show that the AuNPs–PEG sample had the highest drug loading capacity, followed by the AuNPs–silica sample and finally the AuNPs sample. It is worth noting that the AuNPs–PEG sample had a higher drug loading capacity compared to the AuNPs–silica sample despite the fact that the latter had a larger surface area, which suggests that the PEG coating may have facilitated the adhesion of the drug to the nanostructures.

An important aspect to be considered is that the loading of ibuprofen onto nanoparticles can have a significant impact on the physical and chemical characteristics of the nanoparticles. This is because the drug molecules can alter the surface charge and size of the nanoparticles and can also change the stability of the nanoparticle suspensions. Additionally, the drug molecules can interact with the coating materials, leading to changes in their properties, such as the degree of hydration and stability. In particular, the loading of ibuprofen onto the gold nanoparticles can have a significant impact on their physical and chemical characteristics. Further research is necessary to fully understand the mechanisms behind this relationship, and to elucidate how the amount of the drug loaded affects the stability and size of the nanoparticles. This approach highlights the importance of considering the impact of drug loading on the properties of nanoparticle carriers, as it could have implications for their efficacy and safety in the treatment of various diseases.

## 4. Conclusions

To investigate the drug delivery potential of gold nanoparticles coated with silica and polyethylene glycol (PEG), we synthesized gold nanoparticles using a reduction method and coated them with the two materials. The uncoated and coated nanoparticles were characterized using various techniques, including scanning electron microscopy, dynamic light scattering, and Fourier transform infrared spectroscopy. We then conducted drug loading studies to compare the uncoated gold nanoparticles with the silica- and PEG-coated nanoparticles. The results of these studies showed that the coatings significantly improved the drug loading properties of the nanoparticles, making them promising candidates for further development as targeted and controlled release drug delivery systems. More in vitro studies will be necessary to fully assess the potential of these coated nanoparticles as drug carriers and to optimize their design for specific applications. This work highlights the importance of the surface modification or the functionalization of drug delivery systems for targeted delivery and improved drug loading properties. Further studies are necessary to confirm these results and investigate the mechanisms behind the drug loading capacity of these nanostructures as well as the rate of drug release.

## Figures and Tables

**Figure 1 micromachines-14-00451-f001:**
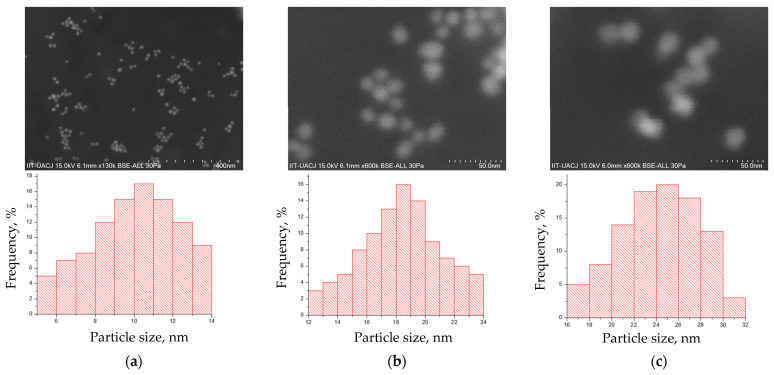
SEM images of the AuNP particles. In panel (**a**), the AuNPs are shown without any coating. In panel (**b**), the AuNPs are coated with PEG, which can be observed to slightly increase the size of the particles and maintain their dispersion. In panel (**c**), the AuNPs are coated with silica, which can be seen to form small aggregates on the surface of the particles. All the particles shown in the figure are spherical in shape.

**Figure 2 micromachines-14-00451-f002:**
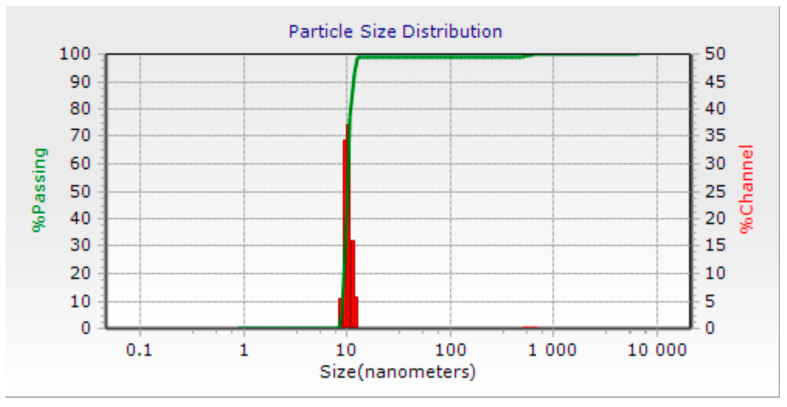
Particle size distribution of the AuNPs. The distribution is narrow and was measured using dynamic light scattering (DLS). The mean particle size of the AuNPs is 10 nm.

**Figure 3 micromachines-14-00451-f003:**
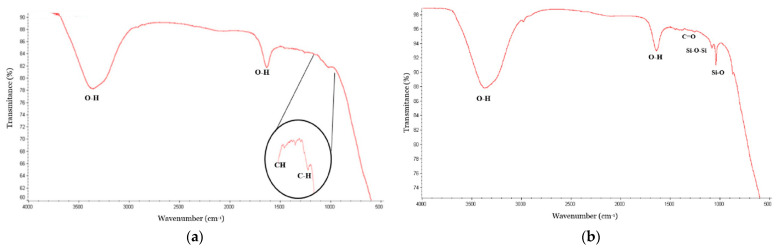
FTIR spectrum of (**a**) the AuNPs coated with PEG, showing the O-H and C-H vibrations bands at 3350 to 3490 cm^−1^ and 1630 to 1750 cm^−1^, which correspond to the O-H flexion and stretching vibrations, respectively, and (**b**) the FTIR spectrum of the AuNPs coated with silica, where the bands at 1100, 1300, 800, and 1000 cm^−1^ correspond to Si-O-Si and Si-O bond vibrations.

**Figure 4 micromachines-14-00451-f004:**
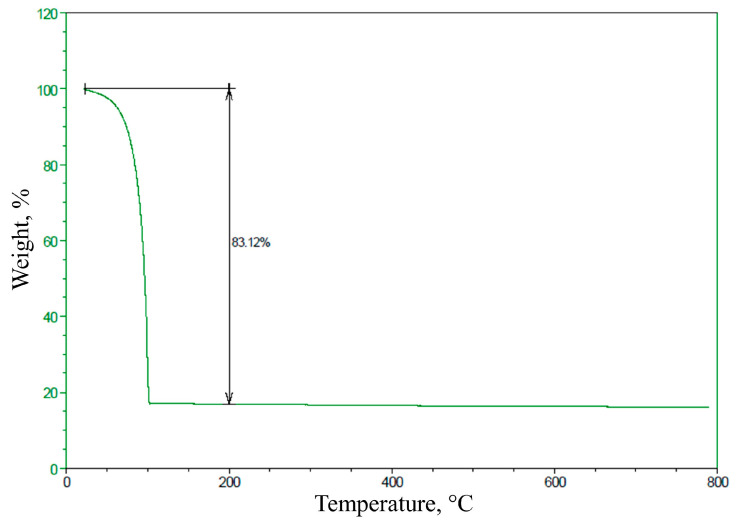
Thermogravimetric analysis of the AuNPs, showing an initial weight loss of 83.12% and a final weight corresponding to the presence of gold in the sample.

**Figure 5 micromachines-14-00451-f005:**
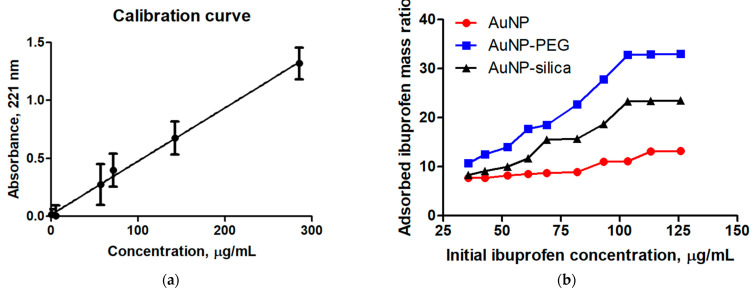
Calibration curve for ibuprofen obtained with triplicate measurements (**a**). In (**b**), the drug loading graphs can be seen, where it is observed that the uncoated AuNPs do not exhibit efficient drug loading, while coating with silica increases the drug loading efficiency, but the PEG coating achieves the best drug loading. The PEG coating achieves a mass loading of 33%, while the silica achieves around 25% when a concentration of 100 micrograms per milliliter of the drug is used, and the uncoated AuNPs achieve around 10%.

## Data Availability

The data presented in this study are openly available in Zenodo at https://doi.org/10.5281/zenodo.7451868 (accessed on 17 December 2022).

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
