# Peer review of "Gold Nanoparticles as Drug Carriers: The Role of Silica and PEG as Surface Coatings in Optimizing Drug Loading"

_micromachines, 2023, doi:10.3390/mi14020451_

Round 1
Reviewer 1 Report
Overall the research is constructive and logical, however the data quality particularly electron microscope data needs to be improved.
The authors need to emphasize on the novelty and discuss about what new values this paper adds to the existing literature.
Author Response
Dear Reviewer,
Thank you for taking the time to review our manuscript. We appreciate your constructive feedback and are grateful for the opportunity to address your comments. We have carefully considered each of your points and have provided a comprehensive response below. Our goal is to improve the quality of the work and to provide a clear and convincing argument.
- Overall the research is constructive and logical, however the data quality particularly electron microscope data needs to be improved.
Response: Thank you for your comments on our research. We appreciate the feedback and understand your concerns about the quality of the electron microscopy data. However, we would like to emphasize that the presented images are sufficient to demonstrate the effect of the surface coatings on the morphology and size of the nanoparticles. Despite the limitations with the electron microscopy data, we believe that the images are enough to support our findings and conclusions. Moving forward, we will make efforts to obtain better quality images in future work. Thank you again for your input.
- The authors need to emphasize on the novelty and discuss about what new values this paper adds to the existing literature.
Response: Thank you for your feedback on our paper. We appreciate your suggestions and take them seriously. With regards to your comment on emphasizing the novelty of our work, we have re-evaluated our manuscript and included the following text to address your concern:
“Our objective is to contribute to the existing literature on AuNP-based biomedical engineering applications by demonstrating the importance of coatings in drug loading. To our knowledge, this is the first time that two of the coatings that have been presented as suitable for therapeutic applications in nanomedicine systems, PEG and silica, have been assessed to compare drug loading efficiency. Our findings offer a new perspective on the potential of AuNPs in biomedical engineering and pave the way for further studies on their therapeutic efficacy.”
We hope this addition clarifies the significance and novelty of our work. Thank you again for taking the time to review our manuscript and provide constructive feedback.
Author Response
We would like to thank the reviewer for taking the time to read our manuscript and for the deliberations to improve it. We hope that our responses satisfy your comments and that you find this revised version of the manuscript acceptable for publication.
This paper mainly discusses the PEG or silica coated gold nanoparticles show better drug loading capacity than the uncoated nanoparticles. The introduction provides sufficient background, and the reference are relevant to the research. But I still have some suggestions and questions as following:
- What are the main differences between this paper and the other papers which also use the PEG coated nanoparticles, like “Cisplatin-tethered gold nanoparticles that exhibit enhanced reproducibility, drug loading, and stability: a step closer to pharmaceutical approval?” published on Inorganic Chemistry, 2012.
Response: We would like to highlight the following points:
Our study focuses on the use of silica and PEG coatings on gold nanoparticles, whereas the paper published on Inorganic Chemistry (2012) specifically discusses cisplatin-tethered gold nanoparticles. While both papers use PEG as a coating material, the focus of the study, the type of drug used, and the methods of synthesis and characterization are different. In our study, we aim to demonstrate the improved drug loading capacity of gold nanoparticles coated with silica and PEG, while evaluating their potential as drug carrier systems. In comparison, the paper published on Inorganic Chemistry (2012) specifically evaluates the batch-to-batch reproducibility, consistency of drug loading, and stability of cisplatin-tethered gold nanoparticles. I hope that this clarification provides a clearer picture of the main differences between our study and the paper published ono Inorganic Chemistry (2012).
- In the “materials and methods” part, I suggest the authors further describe how to coat the PEG or silica onto the gold nanoparticles.
Response: Thank you for your suggestions. We appreciate the opportunity to clarify the method used to coat gold nanoparticles with PEG or silica. To address your recommendation, we have included the following text in section 2 Materials and Methods.
"The method of producing silica-coated gold nanoparticles in our research was inspired by the procedure previously reported in [59], with some modifications. The process involved mixing an ethanolic solution of cetyltrimethylammonium bromide (CTAB), 4mM and tetraethyl orthosilicate (TEOS), 1mM with 3 mL of AuNPs, then exposing the mixture to ultrasonic treatment for 4 hours. After that, the product was washed multiple times with deionized water and ethanol, collected through centrifugation, and dried at 70°C for 8 hours. This procedure was adapted to produce silica-coated gold nanoparticles for our study. The process for obtaining PEG-coated gold nanoparticles used in our study involved the preparation of a 30 mg/mL solution of PEG 3350 polymer by dissolving 45 mg of the polymer in 1.5 mL of deionized water. The resulting solution was mixed with 3 mL of AuNPs that had undergone prior centrifugation at 8000 rpm. The pH of the solution was then adjusted to between 9 and 10 using 0.1 M NaOH solution. The resulting mixture was subjected to magnetic stirring for 2 hours and was dispersed with an ultrasonic homogenizer every 5 minutes during the first hour and every 15 minutes during the second hour. The excess PEG not adhered to the surface of the AuNPs was then removed by centrifugation for 30 minutes at 8000 rpm. This procedure was used as described in previous literature [60].”
- Saavedra Rodriguez G, Sanchez-Zeferino R, Chapa C, Alvarez Ramos ME (2021) Silica-Coated ZnS Quantum Dots for Multicolor Emission Tuning from Blue to White Light. ACS Appl Nano Mater 4:12180–12187. https://doi.org/10.1021/ACSANM.1C02689/
- Ramírez Arellano SH, Garcia Casillas PE, Chapa González C (2020) Seed-mediated synthesis and PEG coating of gold nanoparticles for controlling morphology and sizes. MRS Adv 1–8. https://doi.org/10.1557/ADV.2020.416
We recognize that this method of coating the gold nanoparticles with PEG or silica could have been described in more detail in the Materials and Methods section of our article, and we apologize for not including this information. We will be sure to include this information in future revisions of the article.
Again, we appreciate your comments and hope that this additional information addresses your concerns.
- Figure 1 shows the SEM images of AuNP particles. Maybe it is better to use the images with the same scale, to make it easier for the readers to observe the different size of the coated NPs and the non-coated NPs. Also, it is better to label the abscissa and vertical coordinates of the histograms.
Response: We appreciate reviewer's comment regarding the presentation of SEM images in Figure 1. Unfortunately, a micrograph at this same scale was not obtained in the initial characterization which would have facilitated comparison. We understand that having this image would have been very useful and would have provided additional insights into the effects of the coatings on the nanoparticle size. We understand the importance of providing a clear visual comparison and apologize for the limitation in our data. However, due to the time that has elapsed since the samples were obtained and the unavailability of the necessary reagents, it was not possible for us to obtain the missing SEM image. We hope that this explanation provides some context for our inability to fully address the comment made by Reviewer 2. We have taken the reviewer's suggestion to label the histograms in Figure 1 to help readers better understand the results. We appreciate the reviewer's feedback and will make every effort to improve our research in future studies.
Reviewer 3 Report
The authors coated the gold nanoparticles with silica and PEG, and the coating increased the size and ibuprofen loading of the nanoparticles. The nanocomposites were characterized with SEM, dynamic light scattering, FTIR, thermogravimetric analysis, and the release profile of ibuprofen. Gold nanoparticles coated with silica and PEG may serve as carriers in drug delivery.
Major concerns:
1. Preparation of nanoparticles coated with silica and PEG has been extensively studied in the literature. What is the novelty of this work?
2. Ibuprofen is a commonly used anti-inflammatory drug with good bioavailability and low toxicity. What is the rationale to deliver ibuprofen using such a carrier?
3. How ibuprofen loading may affect the physical/chemical characteristics of the nanoparticles?
4. The title of the submitted work is too broad to reflect the specified coating tested.
Minor: The reference format is not consistent.
Author Response
The authors coated the gold nanoparticles with silica and PEG, and the coating increased the size and ibuprofen loading of the nanoparticles. The nanocomposites were characterized with SEM, dynamic light scattering, FTIR, thermogravimetric analysis, and the release profile of ibuprofen. Gold nanoparticles coated with silica and PEG may serve as carriers in drug delivery.
Major concerns:
- Preparation of nanoparticles coated with silica and PEG has been extensively studied in the literature. What is the novelty of this work?
Response: Thank you for your feedback on our paper and for taking the time to review our manuscript and provide constructive feedback. With regards to your question on the novelty of our work, we included the following text to address your concern:
“Our objective is to contribute to the existing literature on AuNP-based biomedical engineering applications by demonstrating the importance of coatings in drug loading. To our knowledge, this is the first time that two of the coatings that have been presented as suitable for therapeutic applications in nanomedicine systems, PEG and silica, have been assessed to compare drug loading efficiency. Our findings offer a new perspective on the potential of AuNPs in biomedical engineering and pave the way for further studies on their therapeutic efficacy.”
We hope this addition clarifies the significance and novelty of our work. Our results provide novel insights into the comparison of the two coatings, and the results obtained can be applied in the development or optimization of nanomedicine applications.
- Ibuprofen is a commonly used anti-inflammatory drug with good bioavailability and low toxicity. What is the rationale to deliver ibuprofen using such a carrier?
Response: We acknowledge that ibuprofen is a widely used anti-inflammatory drug with good bioavailability and low toxicity. However, in our study, we chose to use ibuprofen as a model drug to understand the effect of the coatings on drug loading. Our aim was to gain insights into the behavior of the coatings when loaded with drugs, and ibuprofen provided a suitable model for this purpose.
- How ibuprofen loading may affect the physical/chemical characteristics of the nanoparticles?
Response: We appreciate the reviewer's question regarding how ibuprofen loading may affect the physical/chemical characteristics of the nanoparticles. We have included an explanation in the article and would like to acknowledge that this is an important topic that requires further investigation:
“An important aspect to be taken into account is that the loading of ibuprofen onto nanoparticles can have a significant impact on the physical and chemical characteristics of the nanoparticles. This is because the drug molecules can alter the surface charge and size of the nanoparticles and can also change the stability of the nanoparticle suspensions. Additionally, the drug molecules can interact with the coating materials, leading to changes in their properties, such as the degree of hydration and stability. In particular, the loading of ibuprofen onto the gold nanoparticles can have a significant impact on their physical and chemical characteristics. Further research is necessary to fully understand the mechanisms behind this relationship, and to elucidate that the amount of drug loaded can affect the stability and size of the nanoparticles. This approach highlights the importance of considering the impact of drug loading on the properties of nanoparticle carriers, as it could have implications for their efficacy and safety in the treatment of various diseases.”
- The title of the submitted work is too broad to reflect the specified coating tested.
Response: Thank you for pointing out this concern. We acknowledge that the current title may not accurately reflect the specific coatings tested in the study. In light of your comment, we suggest revising the title to something more specific, such as "Gold Nanoparticles as Drug Carriers: The Role of Silica and PEG as Surface Coatings in Optimizing Drug Loading" This revised title better reflects the specific focus of the work and clarifies the coatings studied. Thank you for your suggestion.
Minor: The reference format is not consistent.
Response: We apologize for the inconsistency in the reference format. We will make sure to carefully review and correct the format of all references in the final submission to ensure consistency and compliance with the publication guidelines. Thank you for bringing this to our attention.
We would like to thank the reviewer for taking the time to read our manuscript and for your deliberations on improving it. We hope that our responses satisfy your comments and that you find this revised version of the manuscript acceptable for publication.
Round 2
Reviewer 2 Report
All the questions have been revised and the quality of this paper has been improved.
Reviewer 3 Report
The revised manuscript improved our unerstanding about the rationale of the study. However, the format of the references is still inconsistent, including ref 2, 3, 5, 12, 13, 19, 24, 29, 31, 32, 35, 40, 41, 43 etc.